# Epigenetic Alterations of Brain Non-Neuronal Cells in Major Mental Diseases

**DOI:** 10.3390/genes14040896

**Published:** 2023-04-12

**Authors:** Hamid Mostafavi Abdolmaleky, Marian Martin, Jin-Rong Zhou, Sam Thiagalingam

**Affiliations:** 1Department of Medicine (Biomedical Genetics), Boston University Chobanian & Avedisian School of Medicine, Boston, MA 02118, USA; samthia@bu.edu; 2Department of Surgery, Nutrition/Metabolism Laboratory, Beth Israel Deaconess Medical Center, Harvard Medical School, Boston, MA 02215, USA; 3Department of Neurology, Albert Einstein College of Medicine, New York, NY 10461, USA; 4Department of Pathology & Laboratory Medicine, Boston University Chobanian & Avedisian School of Medicine, Boston, MA 02118, USA

**Keywords:** epigenetic, DNA methylation, glia, astrocyte, microbiome

## Abstract

The tissue-specific expression and epigenetic dysregulation of many genes in cells derived from the postmortem brains of patients have been reported to provide a fundamental biological framework for major mental diseases such as autism, schizophrenia, bipolar disorder, and major depression. However, until recently, the impact of non-neuronal brain cells, which arises due to cell-type-specific alterations, has not been adequately scrutinized; this is because of the absence of techniques that directly evaluate their functionality. With the emergence of single-cell technologies, such as RNA sequencing (RNA-seq) and other novel techniques, various studies have now started to uncover the cell-type-specific expression and DNA methylation regulation of many genes (e.g., *TREM2*, *MECP2*, *SLC1A2*, *TGFB2*, *NTRK2*, *S100B*, *KCNJ10,* and *HMGB1*, and several complement genes such as *C1q*, *C3*, *C3R*, and *C4*) in the non-neuronal brain cells involved in the pathogenesis of mental diseases. Additionally, several lines of experimental evidence indicate that inflammation and inflammation-induced oxidative stress, as well as many insidious/latent infectious elements including the gut microbiome, alter the expression status and the epigenetic landscapes of brain non-neuronal cells. Here, we present supporting evidence highlighting the importance of the contribution of the brain’s non-neuronal cells (in particular, microglia and different types of astrocytes) in the pathogenesis of mental diseases. Furthermore, we also address the potential impacts of the gut microbiome in the dysfunction of enteric and brain glia, as well as astrocytes, which, in turn, may affect neuronal functions in mental disorders. Finally, we present evidence that supports that microbiota transplantations from the affected individuals or mice provoke the corresponding disease-like behavior in the recipient mice, while specific bacterial species may have beneficial effects.

## 1. Introduction

The expression and epigenetic dysregulation of hundreds of genes have been shown to be present in mental diseases. While most of the relevant studies have shown expression and different types of epigenetic alterations in blood cells or DNA derived from saliva, more than two dozen studies focus on postmortem brain samples. Since each tissue and each cell type (depending on the age of the tissue) has its own specific gene expression pattern or epigenetic landscape, scrutinizing cell-specific gene expression and epigenetic alterations has remained a challenging undertaking when examining affected postmortem brain samples. Whereas the cerebral cortex has six layers of neurons with different functions, and thus different patterns of gene expression and a different epigenetic landscape, non-neuronal cells comprise a large portion (~50%) of the brain mass and cell count [1,2]. Although for a long time the astroglia cells were thought to provide passive structural support, recent data indicate that they are key regulators of neuronal protection, synaptic pruning and maintenance, and network formation, and hence influence memory and learning as well as cognitive functions. As neuronal cells have different subtypes [3], as shown in Figure 1, several studies have addressed the epigenetic dysregulation of brain neuronal cells in major mental diseases [4,5,6,7,8]. Recently, it has become increasingly clear that a significant degree of the gene expression patterns and associated epigenetic alterations that define mental diseases originate from the brain’s non-neuronal cells, such as the microglia and astrocytes. Glial cells are also present and active in the peripheral nervous system. For example, enteric glia cells interact with the neighboring gut cells and are responsive to enteric bacterial infections and inflammatory factors [9].

Until recently, the isolation of astroglia from neurons and other brain cells for separate expression or epigenetic analysis was a methodological challenge; however, novel techniques, such as single cell RNA-Seq and DNA methylation profiling, may enable comprehensive single cell expression along with epigenetic analyses. Additionally, cell sorting techniques (e.g., using CD11b magnetic beads, which comprise cell-lineage-specific antibodies for microglia isolation) could improve the isolation of specific cells from multicellular tissues [10,11]. Using these techniques now, in the era of multi-omics, scientists can employ correlational gene or protein expression and genetic or epigenetic analyses of different brain regions to identify the main cells involved in disease pathogenesis in order to design novel therapies or preventive remedies for mental diseases. However, it is important to note that, despite the fact that single cell omics could allow for the detection of rare cell populations and the characterization of a diversity of cell types based on gene expression patterns and epigenetic marks, it still lacks the ability to differentiate transient and stable modifications.

In this narrative review, we discuss the gene expression and epigenetic data of glial cells and astrocytes, the primary examples of brain non-neuronal cells, to illustrate their contribution to the pathogenesis of mental diseases. While several epigenetic mechanisms such as DNA methylation, histone modifications, microRNAs, and RNA editing are involved in gene expression regulation in mental diseases, as discussed elsewhere [12,13], our main focus is on DNA methylation alterations; these constitute the main feature of the epigenetic dysregulation of non-neuronal cells, as addressed in the following sections.

## 2. Non-Neuronal Cells of the Brain

In the human brain (regardless of blood vessels and circulating blood cells), different types of non-neuronal cells (Figure 1), including astrocytes, microglia, the progenitor NG2-glia (collectively known as astroglia), and oligodendrocytes, make up almost 50% of the brain’s cells and mass [1,2,14,15]. Oligodendrocytes are involved in the myelination of neuronal cells; meanwhile, astrocytes (derived from the neuroectoderm) are tortuously connected to neuronal synapses and are involved in synapse pruning, among other functions [16]. The progenitor NG2-glia exhibit stem-cell-like behavior (which can generate oligodendrocytes, astrocytes, or neurons to repair brain injury), and, as the brain’s macrophages (which originate from the mesoderm and enter the brain during early embryogenesis), microglia mediate immune responses and clear debris, dead neurons, and infectious elements by phagocytosis. Nevertheless, overactivation of the microglia leads to brain damage in specific CNS diseases [17].

Astrocytes are the most abundant brain non-neuronal cells and have at least four morphological subtypes in human brain [18], including: (i) interlaminar astrocytes, which reside in layer 1 and whose millimeter-long processes extend to layers 2–4; (ii) protoplasmic astrocytes, the most abundant astrocytes in the human brain, which are more complexly arborized compared to rodents and can each cover 250,000–2000,000 synapses; (iii) varicose projection astrocytes that are engaged with the blood vessels; and (iv) fibrous astrocytes, which are larger in size, less branched, and reside in the brain’s white matter.

It appears that the complex development of astroglia is parallel with or secondary to the evolution and expansion of neuronal cells in different species [19]. As reviewed by Vasile and colleagues and illustrated in Figure 2, the glia-to-neuron ratio increases according to the species’ evolutionary stage and cognitive capability. For example, in *C. elegans*, this ratio is 0.18, in rats it is 0.4, in the whole human brain it is 1, and in the human cerebral cortex it is 1.4 [20]. Whereas the structure of the astroglia is more complex in humans, the glia-to-neuron ratio is approximately 25% more than in non-human primates [21]. This implies that astroglia may have more significant roles in functional and cognitive evolution, rather than being merely for structural support. In fact, astrocytes with thousands of processes interact with all of the cell types of the central nervous system (CNS) and are involved in a wide range of functions, supporting the CNS structure, metabolism, blood–brain barrier formation, control of vascular blood flow, axon guidance, synapse formation, and the modulation of synaptic transmission [22]. Although astrocytes are entangled with most synaptic structures, this entanglement is dynamic and rapidly retracts during synaptic expansion but is reestablished after synapse reorganization [23]. Therefore, a continuous and sophisticated neuron–glia interaction is required to preserve the healthy architecture of the adult brain and its function. Hence, any alteration in astroglia characteristics (e.g., de novo mutation or epigenetic alterations) may affect the tailoring of the astroglia–neuron interplay and cause re-adjustments that lead to a neurological or mental disease. Remarkably, while neuronal duplication is rare in adulthood, glia and astrocytes duplicate in the adult brain [24,25]. More importantly, astrocytes can be converted to neurons, which provides an opportunity for neurogenesis derived from astrocytes in adulthood. This is mediated by the PAX6 gene, which is inhibited by miR-365, and its expression is reduced by long-term risperidone treatment [26,27].

Astrocytes also express nearly all neuronal receptors, though their expression depends on the main neurotransmitters released by the neighboring neurons [28]. This indicates that neuronal signaling may synchronize astrocytes for proper activities/functions, such as neurotransmitter removal or ion homeostasis. As ~100 billion human neuronal cells produce 15 trillion electrical and chemical synapses and each neuron uses ~4.7 billion ATP molecules per second, astroglia are among the key providers of ATP and other metabolic supports for neurons and their synapses [19]. Moreover, metabolites, such as lactate, which are produced by astroglia, as well as exogenous L-lactate, induce axon regeneration by acting on metabotropic GABAB receptors (GABBR) expressed by neurons [29]. On the other hand, since the excitatory neurotransmitter glutamate is a strong endogenous neurotoxin and generates plenty of reactive oxygen species (ROS) during neuronal activity, astroglia cells are responsible for the clearance of these endogenous toxins. At the same time, like inhibitory neurons, microglia, i.e., the brain’s immune cells, suppress neuronal activity by catabolizing extracellular ATP (released by astrocytes and neurons upon neuronal activation) to adenosine, which acts on A_1_ receptors, mitigating excessive neuronal activation; meanwhile, microglia ablation induces seizures [30].

Microglia have relatively long lifespans, though shorter than those of neurons. In mice, the lifetime of microglia is around 15 months; thus, half of the microglia population survive a mouse’s lifespan under normal conditions [11,31]. In humans, the median rate of microglia turnover is almost 28% per year, but the lifespan of some microglia could be more than two decades [32]. However, in the event of any brain damage, they have the capacity to proliferate and repopulate in situ [11].

Microglia exert multiple functional roles and contribute to the building of the neuronal circuit through synaptic pruning and stripping during development; they participate in surveillance by secreting neurotrophic factors that react against infectious agents or toxic elements and engage in phagocytic debris clearance, including the removal of dying neurons [33]. In particular, as respiratory, oral, and gut microbiota and their byproducts may pass through mucosal layers and affect different tissues [34], microglia are key players in safeguarding the brain against intrusive infectious and inflammatory elements. However, their overactivity may disturb other brain cells and alter their epigenetic landscapes [17,35]. Here, we provide data regarding glial dysfunction, focusing in particular on epigenetic aberrations in major mental diseases; then, we present supporting evidence indicating that glia dysfunction might be linked to gut microbiome alterations in these diseases.

## 3. Glia Dysfunction in Autism

The role of glia dysfunction, particularly Bergmann Glia in glutamate removal, is well described in autism [36]. Single-cell RNA sequencing revealed that autism-associated transcriptome alterations in specific cortical cell types are related to “synaptic signaling of upper-layer excitatory neurons” and microglia [37]. A large whole-genome study of postmortem brain samples also indicated that DNA methylation alterations associated with autism are involved in the immune system, synaptic signaling, and neuronal regulation and are highly correlated with the affected genes in patients with chromosome 15q duplication and H3K27 acetylation [38].

Some important microglia genes, such as *TREM2* (as the microglia innate immune receptor gene involved in synapse pruning) are also linked to autism pathogenesis. In mice, the lack of expression of *TREM2* is associated with autism-like behavior and, in humans, a reduced *TREM2* protein level correlates with the severity of autism symptoms [39]. Additionally, decreased expression of *TREM2* is associated with increased expression of *TNFA*, a pro-inflammatory cytokine, and *NOS2* (nitric oxide synthase 2) in mice [40]. Interestingly, sodium valproate (an epigenetic drug that inhibits HDACs) decreases *TNF-α* and *NOS2* expression levels [41], hinting at an opportunity for autism epigenetic therapy using HDAC inhibitors. Experimental evidence indicates that *TREM2* is also regulated by microRNAs. In this regard, as it is known that the up-regulation of miRNA-34a (an NF-κB-sensitive miRNA) targets *TREM2* and down-regulates its expression in microglia cells [42], increased expression of miRNA34 a/b/c was also shown in cortical tubers of patients with tuberous sclerosis, an autism spectrum disease [43]. There is also evidence that *TREM2* expression is regulated by DNA methylation. For example, DNA hypomethylation of *TREM2* intron 1, which is associated with its increased expression, was shown in the blood cells of patients with SCZ and Alzheimer’s disease [44,45]. On the other hand, increased DNA methylation of CpG sites located upstream of the *TREM2* transcription start site is reported in the superior temporal gyrus of patients with Alzheimer’s disease [46]. However, in the hippocampus of patients with Alzheimer’s disease, the higher levels of DNA methylation were reported to be due to the enrichment of 5-hydroxymethycytosine associated with upregulation of *TREM2* expression [47]. Considering these data, further study of the epigenetic dysregulation of *TREM2* is warranted in autism.

Methyl-CpG binding protein 2 (*MECP2*) is another important gene in the pathogenesis of autism spectrum syndrome, specifically in Rett syndrome. In general, Rett syndrome is due to the mutation of *MECP2* located in chromosome X. The disease appears mostly in females, as males affected by this mutation usually die shortly after birth. In addition to its mutation, promoter DNA hypermethylation of *MECP2*, associated with its reduced protein expression, was shown in the frontal cortex of male autistic patients [48]. Based on recent data, while neuronal *MECP2* expression is more than that observed in astrocytes, in males, a higher DNA methylation level of *MECP2* regulatory regions is associated with reduced expression of *MECP2* in astrocytes [49]. This supports the idea that astrocytic DNA hypermethylation of *MECP2* may be a mechanism for disease pathogenesis in male autistic patients. In this regard, previous animal studies have shown that the re-expression of astrocytic *MECP2* in globally *MECP2*-deficient mice improves their behavioral and molecular aberrations [50]. Furthermore, as microglia pathology due to *MECP2* dysfunction was later proposed as the leading cause of Rett syndrome and autism pathogenesis [51], it has been shown that MECP2 regulates the expression of “microglia genes in response to inflammatory stimuli” [52].

With the involvement of microglia, it is not surprising that the immune system and complement proteins, such as C1q, C3, and C4, as well as TGFB2, which contribute to synapse pruning during brain maturation [53], are among the key players in autism pathogenesis [54,55] and in other major mental diseases, such as SCZ [56,57]. Relatedly, whole-genome DNA methylation analysis uncovered epigenetic dysregulation of several complement genes such as *C1Q, C3,* and *ITGB2* (*C3R*), as well as several other inflammatory genes (e.g., *TNF-α*, *IRF8*, and *SPI1*) in postmortem brain samples of patients with autism [58]. Therefore, these findings (as summarized in Table 1) call for more studies on the astroglia-mediated epigenetic dysregulation of complement genes in autism.

## 4. Glia Dysfunction in Schizophrenia and Bipolar Disorder

Several lines of evidence indicate that inflammation and inflammation-induced oxidative stress, as well as many insidious/dormant infectious elements, alter the expression status and epigenetic landscapes of brain cells; this evidence is reviewed here and elsewhere [34,83]. One of the best examples of this phenomenon concerns the impact of maternal immune activation on brain cells’ (and, in particular, microglia) gene expression and epigenetic status in conjunction with the pathogenesis of mental diseases [84], as discussed in more detail in the following sections. In addition, analyses of the brain gene expression data in publicly available datasets reveal expression alterations of genes related to cortical astrocytes both in SCZ and in bipolar disorder (BD) [85]. Other human postmortem brain studies also revealed that the altered expression of genes that are important to glia or astrocyte functions (e.g., *SLC1A2* and *TGFB2*) is linked to psychiatric phenotypes [64]. Interestingly, as the expression of astrocytes’ glutamate transporter, *GLT-1* (*SLC1A2*) exhibits >100% and 70% increases in the postmortem brains of patients with SCZ and psychotic BD, respectively [64]. The use of ceftriaxone (an antibiotic that selectively enhances *GLT-1* expression) could reduce prepulse inhibition (which is also reduced in SCZ patients) in rats, which could be reversed by dihydrokainate (DHK), an antagonist of *GLT-1* [86]. Other research findings indicate that *GLT-1* expression is regulated by diverse epigenetic mechanisms [65,66]. For instance, while miR-218 downregulates astrocytic *GLT-1* expression [67] and the hypo-expression of miR-218 increases susceptibility to stress, its reduced expression has been observed in the medial prefrontal cortices of patients with depression and suicide [87,88]. Regarding *TGFB2*, while its expression is increased in the postmortem brains of patients with SCZ and psychotic BD, due to its promoter DNA hypomethylation [64], other studies have shown that *TGFB2* is over-expressed in the neurons of patients with Alzheimer’s disease [89,90]. It is also the only cytokine that is increased in the cerebrospinal fluid of these patients [91]. In vitro studies indicate that the expression of *TGFB2* is induced by toxic amyloid betas in both glial and neuronal cells. In turn, the increased *TGFB2* binds to the extracellular domain of amyloid beta precursor protein and triggers a neuronal cell death pathway in Alzheimer’s disease. Interestingly, the degrees of *TGFB2*-induced cell death are larger in cells expressing a familial AD-related mutant *APP* than in those expressing wild-type *APP* [91,92]. Together, these data suggest the potential roles of *GLT-1* and *TGFB2* epigenetic alterations in the pathogenesis of neuropsychiatric diseases, indicating that they are legitimate targets for therapeutic interventions [93].

Other genes that are mainly expressed by astrocytes and glial cells (e.g., *S100B*, the S100 calcium-binding protein B) are also linked to SCZ pathogenesis in GWAS analysis. Moreover, just as a higher level of the S100B protein is reported in the blood cells of SCZ patients [70], an increased serum level of *S100B* was also reported in BD patients [94]. While *S100B* promotes hippocampal synaptogenesis after traumatic brain injury [95], there is experimental evidence that its expression is regulated by DNA methylation [71].

Another line of evidence in support of the role of astroglia in SCZ is the existence of D2-like receptors in astrocytes. While astroglia account for almost one-third of *DRD2* binding sites in the brain cortex, and *DRD3* is also expressed in astrocytes, mice deficient for this D2-like receptor or that are treated with a DRD3 antagonist do not show astroglia inflammatory activity in response to LPS (lipopolysaccharide) challenge. It should be noted that, although microglia do not express *DRD3*, in *DRD3* deficient mice, the expression of *Fizz1*, an anti-inflammatory protein, is increased in glial cells (both in vitro and in vivo). This also attenuates microglial activation in response to LPS challenge [96]. The fact that commonly used antipsychotic drugs block DRD2-like receptors, and that the long-term use of olanzapine alters *DRD2* promoter DNA methylation levels [97], suggests that the effects of DRD2-like antagonists in SCZ treatment could be due to the inhibition of astroglia’s inflammatory activity, mediated in part by *DRD2* epigenetic modifications.

Human major histocompatibility complex (*MHC*) genes are among other genes associated with microglia functions that are involved in SCZ pathogenesis in GWAS analyses [98,99]. MHC class I is involved in complement-mediated synaptic pruning [38] and exhibits reduced expression in the brains of SCZ patients [72]. Additionally, it has been shown that glia overactivity mediated by complement *C4A* (one of the genes of MHC III) and the increased expression of *C4A* may have deleterious effects in SCZ [62,100]. Notably, in a study of humanized glial chimeric mice, it was shown that mice with glial cells produced from the iPSC of patients with childhood-onset SCZ, exhibited premature glia migration into the cortex and reduced expansion of white matter and its hypomyelination compared to the mice with glia from the normal controls. This was associated with a delay in astrocytic differentiation and abnormality in astrocytic morphology, as well as reduced prepulse inhibition, increased anxiety, and sleep problems. Additionally, the cultured glial progenitor cells from SCZ patients exhibited aberrant expression of genes linked to glial differentiation as well as synapse-associated genes in the RNA-seq analysis, suggesting that the observed glial pathology originates from these cells [101]. An exaggerated synapse pruning was repeatedly reported in adolescents, particularly in SCZ patients [102], which could be mitigated by minocycline [103,104]; meanwhile, it has been shown that the inhibition of microglia activity by minocycline is effective in the treatment of negative symptoms of SCZ in randomized double-blind studies [105,106].

In addition to the relation between genetic variations of the complement system and SCZ [100], there is also evidence that non-genetic alterations of the activity of complement system are associated with SCZ. For example, as summarized in Table 1, increased *C4* and *C1q* levels were reported in the prefrontal cortices of patients with SCZ [59] and the blood cells of antipsychotic-naive first-episode SCZ patients [63], as well as those with chronic SCZ and in individuals at high risk for psychosis; meanwhile, increased *C3* levels were also shown in the latter group [107]. There are also reports of increased levels of *C3a, C5a,* and *C5b-9* in drug-free patients with bipolar disorder [108], and of increased expression of *C1q, C4*, and factor B in the peripheral blood mononuclear cells of chronic BD patients [109]. Epigenetic analysis of different elements of the complement system in other mental diseases revealed those genes of the complement system that are linked to glial activity and are subjects of epigenetic dysregulation (Table 1). For instance, in whole-genome DNA methylation analysis, the epigenetic dysregulation of *C1q, C3,* and *ITGB2* (*C3R*) was reported in autism [58]. The DNA hypomethylation of *C3* associated with its increased expression was also shown in the postmortem brains of patients with Alzheimer’s disease [60,61]. Furthermore, DNA methylation alterations affecting *C4A* and *C4B* expression were reported in a genome-wide DNA methylation analysis of patients with Attention-Deficit/Hyperactivity Disorder [110].

As SCZ and autism are more common in males, it is important to note that, in females, one of the X chromosomes is subject to random inactivation by DNA methylation. Hence, if the activity of any gene in one of the X chromosomes is imbalanced due to inherited or de novo mutations, in a female subject, half of the neighboring cells can work normally, partially balancing the tissue functions. For example, *SRPX2*, which is in chromosome X and is involved in language and cognitive development [111], exhibits expression reduced by almost 20% in the postmortem brains of SCZ patients [64]. Although the *SRPX2* gene codes a neuronal protein, C1q binds to *SRPX2*, inhibiting synapse eliminations [112]. Thus, a close cooperation between *SRPX2* and this complement is required for the fine tuning of synapse pruning in normal brain development. In cancer research, it has been shown that DNA methylation regulates *SRPX2* expression levels [113]. Therefore, DNA methylation alteration of *SRPX2* could be an interesting subject for further studies in SCZ, as well as in autism and dyspraxia, which are both more prevalent in males than in females. There is also a correlation between the expression of *MECP2*, a methyl CpG binding protein, and *SRPX2* expression [114], which warrants further research.

Other evidence related to astroglia epigenetic alterations in mental diseases comes from imprinted genes in which one copy of the parental alleles (in autosomes) is inactivated by DNA methylation. In this regard, whole-genome DNA methylation analysis for rare epigenetic variations identified that the NDN gene, which is highly expressed in astrocytes, was linked to SCZ as well as to autism pathogenesis [73]. This gene is exclusively expressed from the paternal allele and is in the Prader-Willi syndrome deletion region implicated in autism pathogenesis [115].

## 5. Astroglia Pathology and Dysfunction in Depression

In addition to SCZ and BD, there is evidence for astroglia dysfunctions in depression. For example, whole-transcriptome analysis using RNA-seq of human postmortem brain samples from drug-free individuals with MDD (major depressive disorder) and suicide revealed deficits in genes related to microglial and astrocytic cell functions [116]. Aberrant DNA methylation patterns specific to astrocytes were also shown in the prefrontal cortices of postmortem brain samples of patients with depression [117]. Another study reported the upregulation of astroglia’s potassium channel gene (*Kir4.1* or *KCNJ10*) and reduced *GLT-1* (*SLC1A2*) activity (which removes ~90% of extracellular/synapse glutamate) and increased neuronal bursting activity of the lateral habenula as key factors in the induction of depression-like behaviors [68,69]. A recent study revealed that DNA methylation regulates *KCNJ10* expression in astrocytes [74]. Aberrant DNA methylation of *NMDAR* (more specifically, the hypermethylation of the *GRIN2A* subunit) was also reported in the hippocampus and prefrontal cortex of MDD patients [76]. However, in SCZ patients, DNA hypomethylation of *GRIN2B* was shown in blood cells [118]. Interestingly, ketamine, which is used to treat MDD, decreases neuronal bursting activity [119] by blocking glial *NMDAR* in the lateral habenula, which is considered to be the brain’s “antireward” center [68,69]. Nevertheless, in rats, ketamine’s effects on depressive-like behavior was attributed to its activity in the regulation of astrocytic *GLT-1*, and also through BDNF-TrkB signaling [120]. It has also been shown that ketamine alleviates DNA hypermethylation of *BDNF* in the medial prefrontal cortex and hippocampus in a mouse model of PTSD [121]. Furthermore, while *BDNF* and its receptor NTRK2 play key roles in astrocytes’ maturation and functions [122], DNA hypermethylation of *NTRK2* and its reduced expression was reported in the postmortem brains of patients who died by suicide [75].

*HMGB1* is another microglia-associated gene involved in depression [77,123]. Animal studies have shown that unpredictable chronic stress can lead to microglia activation in the hippocampus and depressive-like symptoms [124]. This type of stress could increase *HMGB1* expression in the hippocampal microglia, and the infusion of *HMGB1* into the mice hippocampus could also induce depression [78]. Interestingly, the activation of the microglia, along with depressive symptoms, could be prevented by minocycline or imipramine [124]. While *HMGB1* is a well-known marker of inflammation, increased expression of *HMGB1*, associated with its promoter DNA’s methylation alteration, was reported in cardiac progenitor cells following hypoxia and metabolic diseases [79,80], suggesting that DNA methylation is a mechanism for *HMGB1* regulation. However, in brain cells, *HMGB1* expression is also regulated by HDAC4&5 and miR-129 [81,82], the latter of which was shown to regulate neuronal migration in mice brains [125].

As is the case for SCZ and BD, these studies link epigenetic dysregulations of astroglia to depression (summarized in Table 1), which is attenuated by therapeutic interventions. However, more studies are needed to identify which internal or external factors (other than unpredictable chronic stress) may be responsible for astroglia dysfunction in depressive and other mental disorders, a subject that we touch on in the following section.

## 6. Microglia and Astrocytes, the Ambassadors of Microbiome Communication with the Brain

While glial cells reside in the enteric peripheral nervous system and respond to gut infection/inflammation [9], and the dysfunction of glial cells has been shown in major mental diseases (as described above), an altered gut microbiome has been reported in several psychiatric diseases such as autism [126,127,128], SCZ [129], depression [130,131], and Alzheimer’s disease [132,133]. In addition to the gut, a saliva microbiome analysis of drug-naïve SCZ patients also revealed that the ratio of Firmicutes to Proteobacteria was enriched stepwise from healthy controls to clinically high-risk individuals after the first episode of SCZ [134].

Interestingly, as shown in Figure 3, the transplantation of gut microbiota from patients with autism, SCZ, or MDD to germ-free mice induces corresponding disease-like behaviors [131,135,136]. Similarly, fecal transplantation from old to young mice results in poor performance in spatial learning and memory tests; this is associated with the altered expression of hippocampal proteins linked to synaptic plasticity and neurotransmission. The microglia of the recipient mice also exhibit an ageing-like phenotype in the hippocampus fimbria [137]. Moreover, these behavioral changes are associated with neurotransmitter or metabolic alterations. For example, mice with SCZ symptoms exhibit decreased glutamate and increased glutamine and GABA levels in the hippocampus, and mice with depressive symptoms exhibit alterations in host metabolites linked to the amino-acid and carbohydrate metabolisms when compared to control mice [131,136]. Furthermore, fecal transplantation from mice with stress-induced depression to normal mice can also induce depression in the recipient mice; this is associated with a decrease in endocannabinoid (eCB) signaling as a result of the reduced production of peripheral fatty acids, which are precursors of eCB ligands. However, the adverse effects of altered microbiota can be alleviated by the selective enhancement of “central eCB or by complementation with a strain of the *Lactobacilli* genus” [138]. Other bacteria, such as *Mycoplasmataceae*, were also shown to affect blood *S100B* levels [134], which are altered in SCZ as described above.

Microbiota manipulation in germ-free mice also impacts fear extinction learning, which is associated with gene expression alterations of the glia, excitatory neurons, and other brain cells in the medial prefrontal cortex in single-nucleus RNA sequencing analysis [139]. These mice exhibit postsynaptic dendritic spine remodeling and hypoactivity of cue-encoding neurons, according to transcranial two-photon imaging analysis. The down-regulation of four metabolites linked to neuropsychiatric disorders was also shown in a metabolomic analysis of these germ-free mice, while selective microbiota re-establishment could restore normal extinction learning [139]. On the other hand, the lack of a natural microbiome also affects brain functions. For instance, germ-free mice exhibit increases in plasma tryptophan and hippocampal concentrations of serotonin and its metabolite (5-hydroxyindoleacetic acid) compared to control mice. As the brain’s neurochemical changes remain stable until adulthood, restoring microbial colonization after weaning could reverse behavioral alterations in the affected mice [140].

The science of the microbial colonization of the human gut was limited before the development of microbial 16S rRNA sequencing technology. Now, it has been revealed that the human gut contains thousands of bacterial elements, the total number of which is ten times more than the number of cells in the human body. It is also known that the gut microbial population (the microbiome or microbiota) collectively contains 100 times more genes than humans and makes numerous compounds, nutrients, and vitamins that are required for the epigenetic fine tuning of brain genes’ functions, as reviewed elsewhere [34]. As an example, the gut microbiota produces short-chain fatty acids (SCFAs), which are well-known epigenetic modifiers and protect neurons via free fatty acid receptor 2 (FFAR2) signaling [141].

In contrast to the traditional view that the infant gut microbiota begins to develop after birth following exposure to the mother’s gut microbiota and environmental microbes, recent PCR-based microbiome studies provide strong evidence that the alimentary tracts of newborns are not sterile before birth [142]. In fact, the fetus acquires the mother’s microbiota while in the uterus, most likely from the blood circulation. This means that the mother’s gut bacteria cross the intestinal wall (like viral elements) and, through blood circulation, end up in the fetus’s gut. However, after birth, the microbial community of the mother’s milk (which is affected by several factors such as the maternal diet, pre-pregnancy BMI, and antibiotic use during or after pregnancy) further develops the infant’s gut microbiome community [143,144]. This supports the idea that a significant aspect of the familial transmission of mental disease might relate to the transmission of pathological microbiota (rather than inherited genes) to other family members. In fact, while most of the major psychiatric disorders are believed to be inherited genetic diseases, large-scale genome-wide association studies failed to identify specific genes (or a number of genes) with an effect size of even 1% as being responsible for the pathogenesis of major mental diseases [145,146]. Therefore, the transmission of a pathological gut microbiota to family members, particularly from a mother to her offspring (even before birth), may imitate the futures of genetic diseases.

There is also ample evidence that the respiratory or oral and gut microbiota may pass through the respiratory or intestinal wall and reside in different tissues after birth. Although it is generally presumed that the immune system can ultimately eliminate or inactivate these intrusive elements from the blood or affected tissues, this process may not be complete and could have functional consequences both in the immune system and in the affected tissues. Indeed, there might be borderline states, during which sub-chronic immune activity may be present, accompanied with tissue inflammation and subsequent tissue damage. Examples of such scenarios are well known in settled lung tuberculosis and in chronic salmonella infection of the bile tract, or in the many latent viral and parasitic infections described in medical textbooks (e.g., chronic active hepatitis of the hepatitis B/C viruses, HIV, herpes simplex type 1/2, and parasitic infections such as toxoplasmosis). More recent studies also indicate that almost 50% of the population are carriers of the herpes virus in the cell nucleus of their peripheral nervous systems [147]. In relation to specific brain disease, a larger number of microbial 16S rRNAs was reported in the serum of patients with Parkinson’s disease as compared to the control subjects [148].

Considering the continuous intrusion of infectious elements into the blood circulation, and thus into different tissues, it is not surprising that the brain tissue has an additional immune system, the microglia, which cooperates with astrocytes to secure its protection against infectious elements. While astrocytes are also involved in other functions, such as neuronal nurturing, nutritional support, and synapse pruning, their unbalanced activation, as well as malfunctions that occur due to the impact of infectious (e.g., HIV) or toxic elements and the resultant inflammation, could affect the brain’s neuronal network [149]. This could be limited to a critical developmental period or could be chronic, with specific or non-specific presentations. In this regard, the potential contribution of maternal immune activation involving several cytokines that affect microglia activity is well described in the pathogenesis of autism and SCZ [150,151]. In an animal model of psychiatric disease, it was shown that maternal immune activation in conjunction with stress in the prepubertal maturation period can lead to more severe behavioral symptoms in adulthood, which can be prevented by anti-inflammatory intervention (e.g., minocycline treatment) during stress exposure [152]. Immune activation can be further induced by the synergistic action of specific gut bacteria. For example, it has been shown that a strain of *Lactobacillus reuteri* has a peptide resembling myelin oligodendrocyte glycoprotein. Meanwhile, a strain in the *Erysipelotrichaceae* family could intensify the responses of T helper cells; together, these activate autoreactive T cells in the small intestine, leading to the worsening of symptoms in mice models of multiple sclerosis [153]. There is also evidence that, through T helper cells, specific gut fungi produce IL-17, which affects social behavior in mice through neuronal IL-17 receptor signaling [154].

Other studies have shown that microglia are key elements of the “gut–brain axis” in transmitting the impacts of gut microbiota into the brain. For instance, a recent study reports that “gut microbiota–driven brain Aβ amyloidosis in mice requires microglia” to manifest [155]. On the other hand, treatment with specific bacterial species such as *Clostridium butyricum* could prevent microglial activation and Aβ deposits, which are associated with reductions in inflammatory cytokines and the improvement of cognitive functions in mice models of Alzheimer’s disease [156]. In addition, while maternal microbiome dysbiosis causes neurodevelopmental disorders, associated with the increased expression of microglial senescence genes and synaptic alterations, lactobacillus could rescue microglial activation/dysfunction and postnatal neurobehavioral abnormalities [157]. Interestingly, other types of environmental distress, such as psychological stress, may also influence microglia inflammatory activity and impact synapse pruning mediated by the activity of the peripheral immune system, including helper T cells [158]. However, inflammatory signals may also contribute to microglial maturation [11]. In fact, the movement of helper T cells into the brain tissue (at least around birth) influences microglia maturation and synapse pruning, whereas its inhibition results in an extra number of synapses [159]. Overall, the exact mechanism(s) involved in the epigenetic dysregulation of non-neuronal cells that influence the pathogenesis of neurodegenerative disorders are yet to be delineated. However, the most likely explanation is the crossing of the blood–brain barrier (BBB) by cytokines and metabolites generated by the host’s gut microbiome to regulate the activity of epigenetic modulation enzymes (Figure 4).

Altogether, these lines of evidence clearly suggest that the external or internal borders of the human body are subject to constant invasions and perturbations by infectious elements, and the immune system is responsible for an ongoing clean-up operation. Whenever the quantity of intrusive elements is high or when the functionality of the immune system is compromised or overactivated, the affected individuals may show disease manifestations. Abnormal gut microbiota may induce astroglia inflammation as well as inflammation-induced oxidative stress, which may alter the epigenetic landscapes of microglia or astrocytes, triggering brain pathologies. In these conditions, the impact of chronic inflammatory reactions can be remarkable, especially in inducing epigenetic alterations of the immune system and the affected tissues [34,160]. On the other hand, several lines of clinical evidence support the potential application of microbiome restoration in the treatment of mental diseases [34]. More recently, a functional MRI (fMRI) study demonstrated that four weeks of probiotic administration could alter the brain’s “activation patterns in response to emotional memory” tasks in humans [161]. Other emerging evidence also supports the idea that probiotics are useful in the treatment of autism, depression, and SCZ [162,163,164,165].

## 7. Conclusions and Future Prospects

It is becoming increasingly clear that changes in the microenvironment could introduce functional diversity into non-neuronal cells, causing alterations in the neuronal circuits that could lead to mental health diseases. Transcriptionally divergent non-neuronal cell populations that are responsible for disease and which cause neuronal circuits could emerge due to the effects of factors that are present in or introduced to the microenvironment or due to unique cell–cell interactions. The current data strongly support the notion that astroglia (astrocytes and microglia) dysfunctions due to epigenetic or other alterations are linked to the pathogenesis of major mental diseases mediated by inflammatory mechanisms. While there is ample evidence that the gut microbiome might be an important contributor in astroglia dysfunction and disease pathogenesis (Figure 4), it remains unclear what other internal (e.g., metabolic dysfunction) or external (e.g., nutritional imbalance or toxic components) factors may be responsible for astroglia epigenetic alterations in major mental diseases.

From a therapeutic point of view, unlike neurons, which barely duplicate in the adult brain, astrocytes and glia maintain their capability for duplication [24]. Fortunately, novel techniques also allow for the in situ differentiation of astrocytes to neurons, which repopulate endogenous neural circuits. In fact, astrocytes can be converted to neurons mediated by the *PAX6* gene, which is inhibited by miR-365. This provides an opportunity for neurogenesis from astrocytes in adulthood [27] in specific brain diseases [166].

## Figures and Tables

**Figure 1 genes-14-00896-f001:**
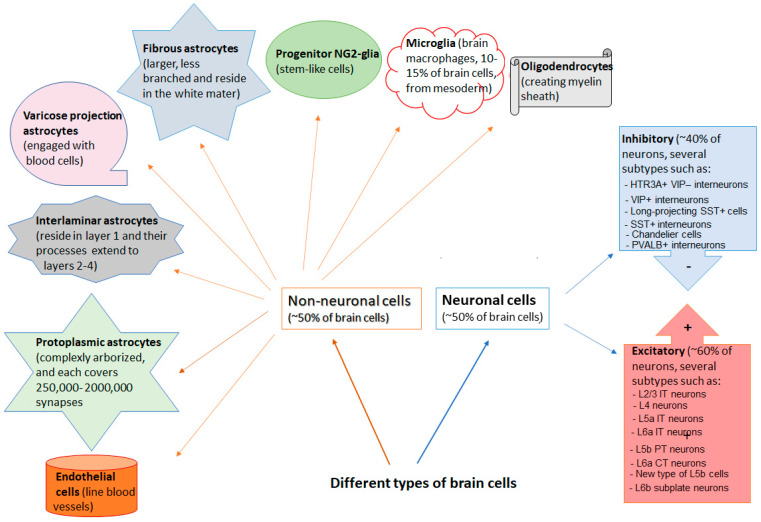
**Composition of different types of brain cells.** Note that only the microglia and endothelial cells originate from the mesoderm, while the other cells originate from the ectoderm.

**Figure 2 genes-14-00896-f002:**
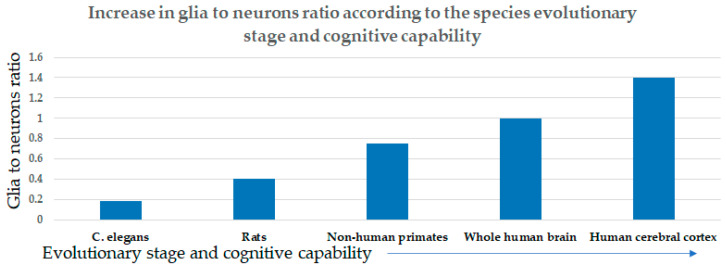
**Glia-to-neurons ratio, according to the species’ evolutionary stage.** The glia-to-neuron ratio increases as the species’ evolutionary stage progresses and its cognitive capability increases.

**Figure 3 genes-14-00896-f003:**
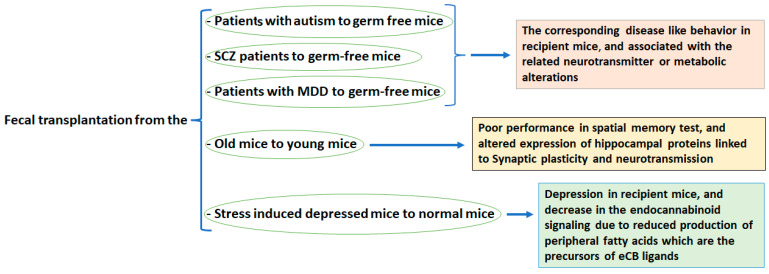
**The proposed link between microbiome alterations and behavioral changes.** Multiple lines of evidence support the finding that microbiota transplantations from the affected individuals or mice provoke the corresponding disease-like behavior in the recipient mice.

**Figure 4 genes-14-00896-f004:**
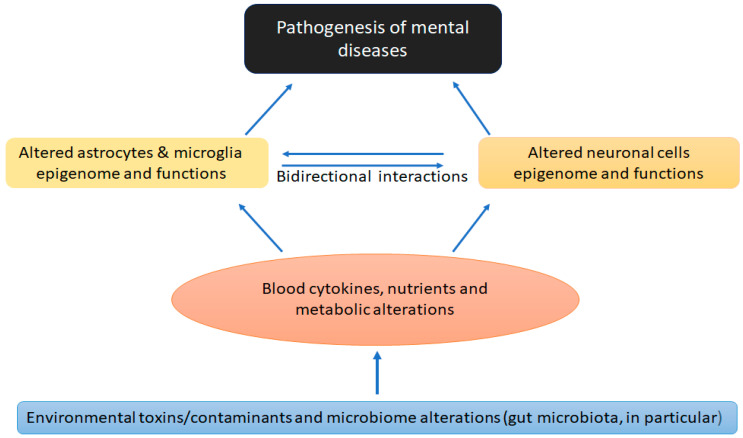
**Environmental impacts on the microbiome and the known mechanisms through which brain functions are affected.** As illustrated, microbiome alterations not only affect blood inflammatory cytokines, which in turn affect brain cell epigenome and functions (see Table 1), but they also affect gut and blood nutrients and metabolites. These then influence neuronal and non-neuronal cells’ epigenetic landscape and functions, leading to disease pathogenies.

**Table 1 genes-14-00896-t001:** Genes linked to non-neuronal brain cell function and supporting evidence indicating their epigenetic dysregulation in mental diseases.

Gene	Active in	Functions	Phenotypes	Expression Status	Epigenetic Alteration(s)	References
*TREM2*	Microglia	synapse pruning	Autism	Decreased in brain	Increased miRNA-34a	[39,43]
			Alzheimer’s Disease	Increased in blood cells	DNA hypo-methylation	[44]
			SCZ	Increased in blood cells	DNA hypo-methylation	[45]
				Superior temporal gyrus (No expression study)	DNA hyper-methylation	[46]
				Increased in hippocampus	DNA hypermethylation (higher 5-hydroxymethylcytosine)	[47]
*MECP2*	Astrocytes	Neurodevelopment and regulation of microglia genes	Autism	Reduced in the frontal cortex	DNA hyper-methylation	[48,49]
*C1q*, *C3* and *C4*	Microglia and astrocytes	Synapse pruning	Autism and SCZ	Increased in several brain areas (e.g., DLPFC)	Different DNA methylation alterations	[54,55,56,57,58,59]
*C3*	Microglia	synapse pruning	Alzheimer’s Disease	Increased in brain and middle temporal gyrus	DNA hypo-methylation	[60,61]
*C4a*	Microglia	synapse pruning	SCZ	Increased in blood cells	Not defined in SCZ (regulated by DNA methylation in ADHD)	[62,63]
*SLC1A2*/*GLT1*	Astrocytes	Glutamate transporter and extracellular synapse glutamate removal	SCZ, BD	Increased in brain	Regulated by miR-218, DNA methylation and histone acetylation	[64,65,66,67]
			Depression	Reduced in lateral habenula	?	[68,69]
*S100B*	Mainly astrocytes	Hippocampal synaptogenesis	SCZ	Increased in blood cells and serum	DNA methylation alterations	[70,71]
*MHC class I*	Microglia	Synaptic pruning	SCZ	Reduced in brain (DLPFC) and blood	DNA methylation alterations	[58,72]
*NDN*	Astrocytes	Neurodevelopment, spine formation	SCZ and autism	?	DNA Hypo-methylation (Imprinted gene)	[73]
*KCNJ10*	Astrocytes	A potassium channel	Depressive symptoms	Increased in lateral habenula	Regulated by DNA methylation	[69,74]
*NTRK2*	Astrocytes	Astrocyte maturation	Suicide	Decreased in brain	DNA hyper-methylation	[75]
			SCZ	Increased in DLPFC	?	[64]
*GRIN2A*	Astrocytes	Aβ cleanup	Depression	?	DNA hypermethylation	[76]
*HMGB1*	Microglia	Inflammation. stimulates microglia	Depression	Increased in hippocampal microglia and serum	Regulated by DNA methylation, HDAC4&5, and miR-129	[77,78,79,80,81,82]

## Data Availability

Not applicable.

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
