# Peer review of "Epigenetic Alterations of Brain Non-Neuronal Cells in Major Mental Diseases"

_genes, 2023, doi:10.3390/genes14040896_

Round 1
Reviewer 1 Report
The manuscript of Abdolmaleky et al. presents the current state of knowledge on the expression of genes using scRNA-seq in non-neuronal-brain cells. The introduction gives background on the types of non-neuronal brain cells and the text is well-illustrated with figures. The results describe in details the gene involved and their possible epigenetic regulations. In this part, although in is comprehensive, I would try to generalize and include details in tables. There are also other aspects, such as gut microbiome mentioned, although they do not relate directly to the main topic. In conclusion, there are some important aspects highlighted, but the technical aspects of omics techniques I would rather describe in the introduction or limitations, not in the conclusion.
Comments:
While non-neuronal cells are important, there are also several studies on the importance of epigenetic regulation of neuronal cells. Please discuss.
I assume that it is a narrative review. If so, please state it. If not please provide the search criteria for manuscripts.
Please state clearly which studies apply to humans and which for an animal model
In the Introduction, I would also explain what is meant by epigenetic regulation (methylation, miR, others?)
I find table 1 very helpful. For clarity, I would suggest doing similar tables for other diseases mentioned.
The paragraph with a gut microbiome, although interesting, does not relate very well to the title of epigenetic alterations. Please elaborate on this link.
Please add limitations of the study and of the scRNA-Seq technique
I would significantly shorten conclusions.
Minor comments:
Please write human genes in italics and check if there is a proper nomenclature for mice genes, proteins etc.
Author Response
Comments and Suggestions for Authors
The manuscript of Abdolmaleky et al. presents the current state of knowledge on the expression of genes using scRNA-seq in non-neuronal-brain cells. The introduction gives background on the types of non-neuronal brain cells and the text is well-illustrated with figures. The results describe in details the gene involved and their possible epigenetic regulations. In this part, although in is comprehensive, I would try to generalize and include details in tables. There are also other aspects, such as gut microbiome mentioned, although they do not relate directly to the main topic. In conclusion, there are some important aspects highlighted, but the technical aspects of omics techniques I would rather describe in the introduction or limitations, not in the conclusion.
Response: We thank the reviewer for constructive comments. New data and studies have been included in Table 1 and “technical aspects of omics techniques” was transferred to the introduction from the conclusion.
Comments:
- While non-neuronal cells are important, there are also several studies on the importance of epigenetic regulation of neuronal cells. Please discuss.
Response: New citations addressing epigenetic alterations of neuronal cells now are now included in the introduction of MS.
- I assume that it is a narrative review. If so, please state it. If not please provide the search criteria for manuscripts.
Response: In the introduction, we mention that this work is a narrative review.
- Please state clearly which studies apply to humans and which for an animal model
Response: This has been clarified in the text and the related figures.
- In the Introduction, I would also explain what is meant by epigenetic regulation (methylation, miR, others?)
Response: We added a sentence in the introduction in response to this comment.
- I find table 1 very helpful. For clarity, I would suggest doing similar tables for other diseases mentioned.
Response: Table 1 (now modified) includes data related to all diseases (schizophrenia, bipolar disorder, autism, depression, and Alzheimer’s diseases) that are discussed in this review.
- The paragraph with a gut microbiome, although interesting, does not relate very well to the title of epigenetic alterations. Please elaborate on this link.
Response: We added a few sentences in the text to address this issue (page 13). However, if desirable we are open to modifying the title to “Epigenetic alterations of brain non-neuronal cells in major mental diseases: potential involvement of gut microbiome” to address the link in the title.
- Please add limitations of the study and of the scRNA-Seq technique
Response: A sentence in response to this comment has been added to the MS on page 2, second paragraph.
- I would significantly shorten conclusions.
Response: A paragraph of the conclusion has been transferred to the introduction (as suggested) and conclusion has been shortened by almost 40%.
Minor comments:
Please write human genes in italics and check if there is a proper nomenclature for mice genes, proteins etc.
Response: Human genes has been written in italic in the entire MS as well as Table 1.
Reviewer 2 Report
This is a well written review on the possible role of epigenetic changes in glial cells in the evolution of several common neuropsychiatric diseases: ASD, Schizophrenia and bipolar disorder. It is an important contribution to our understanding of these disorders. The last part regarding the possible connection of gut microbiome to neuropsychiatric diseases, as evidenced in experimental animal studies also fits the purpose of this review. The conclusions fit well the purpose of this review. I have no comments to the authors.
Author Response
Open Reviewer 2
Comments and Suggestions for Authors
This is a well written review on the possible role of epigenetic changes in glial cells in the evolution of several common neuropsychiatric diseases: ASD, Schizophrenia and bipolar disorder. It is an important contribution to our understanding of these disorders. The last part regarding the possible connection of gut microbiome to neuropsychiatric diseases, as evidenced in experimental animal studies also fits the purpose of this review. The conclusions fit well the purpose of this review. I have no comments to the authors.
Response: We thank the reviewer for the inspiring comments.
Reviewer 3 Report
(See attached file)
Overall this is a wonderful contribution. My main concern is that there are little to no visualizations in such a complex review paper. Figure 1 should be rotated to vertical and include something more intuitive than just text - even simple representations. Table 1 is fine but the caption needs to be on the bottom for consistency and needs a description of the overall message of the table. Figure 2 should also be rotated or the middle ovals need to be modified. This and the first figure are crucial information but hard to read. There are a few more places where I added some quick questions or notes on the attached PDF.

Author Response
Reviewer 3 comments:
Overall this is a wonderful contribution. My main concern is that there are little to no visualizations in such a complex review paper. Figure 1 should be rotated to vertical and include something more intuitive than just text - even simple representations. Table 1 is fine but the caption needs to be on the bottom for consistency and needs a description of the overall message of the table. Figure 2 should also be rotated or the middle ovals need to be modified. This and the first figure are crucial information but hard to read. There are a few more places where I added some quick questions or notes on the attached PDF.
Response: We thank the reviewer for encouraging and constructive comments.
Now Figure 1 has been modified and rotated to the vertical direction. Also, Table 1 is modified and based on the journal method the table caption placed at the top of the Table 1. The middle oval of the Figure 2 (now numbered Figure 3) has been modified to fix the problem during pdf generation of the MS.
In response to notes mentioned in page 3 of the PDF file, a new figure has been generated and now included in the MS (Figure 2 of the revised MS). In response to the first comment on page 5, Table 1 has been modified and in response to the second comment, the text has been modified by adding 50 words and by addressing the details to the following sections. In response to the comment mentioned in page six, unfortunately we could not generate a figure due to the complexity of the subject matter.
In response to the comment mentioned in page 7, we addressed the key points in Table 1, for more clarification.
In response to the comment mentioned in page 8, the key points related to depression are included in Table 1.
In response to comment of page 11, a figure has been generated and included in the MS as figure 4.
Finally, mismatching fonts and missing or extra spaces have been corrected in the revised MS.
Round 2
Reviewer 1 Report
Thank you for the revsions. This is a valuable manuscript on an important topic.